# Quantum Multi-Round Resonant Transition Algorithm

**DOI:** 10.3390/e25010061

**Published:** 2022-12-28

**Authors:** Fan Yang, Xinyu Chen, Dafa Zhao, Shijie Wei, Jingwei Wen, Hefeng Wang, Tao Xin, Guilu Long

**Affiliations:** 1State Key Laboratory of Low-Dimensional Quantum Physics and Department of Physics, Tsinghua University, Beijing 100084, China; 2Beijing Academy of Quantum Information Sciences, Beijing 100193, China; 3Department of Applied Physics, School of Science, Xi’an Jiaotong University, Xi’an 710049, China; 4Shenzhen Institute for Quantum Science and Engineering, Southern University of Science and Technology, Shenzhen 518055, China; 5Tsinghua National Laboratory for Information Science and Technology, Beijing 100084, China; 6Collaborative Innovation Center of Quantum Matter, Beijing 100084, China

**Keywords:** quantum computing, quantum simulation, resonant transitions, nuclear magnetic resonance

## Abstract

Solving the eigenproblems of Hermitian matrices is a significant problem in many fields. The quantum resonant transition (QRT) algorithm has been proposed and demonstrated to solve this problem using quantum devices. To better realize the capabilities of the QRT with recent quantum devices, we improve this algorithm and develop a new procedure to reduce the time complexity. Compared with the original algorithm, it saves one qubit and reduces the complexity with error ϵ from $O(1/\epsilon^{2})$ to O(1/ϵ). Thanks to these optimizations, we can obtain the energy spectrum and ground state of the effective Hamiltonian of the water molecule more accurately and in only 20 percent of the time in a four-qubit processor compared to previous work. More generally, for non-Hermitian matrices, a singular-value decomposition has essential applications in more areas, such as recommendation systems and principal component analysis. The QRT has also been used to prepare singular vectors corresponding to the largest singular values, demonstrating its potential for applications in quantum machine learning.

## 1. Introduction

Calculating the eigenvalues and eigenstates of a Hamiltonian is a fundamental problem in quantum physics and chemistry. Furthermore, the singular-value decomposition (SVD) for non-Hermitian matrices plays a vital role in more fields. Fundamentally, one can solve the eigenequations of the matrices to get this information. However, calculating this problem is ineffective because the computational cost grows exponentially if the dimension of a matrix grows exponentially with the size of the systems, such as a molecular Hamiltonian [1].

Several quantum algorithms have come up to effectively obtain the energy spectrum in a quantum computer. The quantum phase estimation algorithm (PEA) [2,3] is one of the most famous quantum algorithms, which can obtain the eigenvalues of a Hamiltonian with a quantum advantage over classical algorithms. Nevertheless, before using the PEA, we should usually prepare an eigenstate as the initial state. For a complicated system, giving an appropriate guess state before the PEA is difficult. The adiabatic state preparation [4] is another quantum algorithm for preparing an eigenstate of a system, but the efficiency of the ASP algorithm depends on the minimum energy gap between the ground state and the first excited state of the adiabatic evolution Hamiltonian along the evolution path between the initial Hamiltonian and the system Hamiltonian, which is difficult to estimate in practice [5,6]. Many quantum algorithms aim to solve the ground state and energy for a given Hamiltonian. The quantum eigenvalue transformation of unitary matrices with real polynomials (QTE-U) is a useful tool that uses a single ancilla qubit and no multi-qubit gate to estimate the ground-state energy and prepare the ground state [7]. The quantum eigenvalue estimation algorithm can estimate the eigenvalues of a Hamiltonian from the expectation values of the evolution operator for various times [8]. Universal quantum cooling algorithms are another way to prepare the ground state by utilizing a dual-phase representation of decaying functions to universally and deterministically realize a general cooling procedure with shallow quantum circuits [9]. A variational quantum eigensolver [10] is one way that can provide a quantum over-classical advantage to accomplish this task by using a classical computer and a noisy intermediate-scale quantum device with a low coherence time. Almost all algorithms based on the variational principle need many samples to obtain the estimated values of the different items. A full quantum eigensolver [11] based on a linear combination of unitary operators [12,13,14,15,16] has obtained rapid theoretical development [17,18] and some state-of-the-art experimental demonstrations [18,19].

The quantum resonant transition (QRT) algorithm is one way to obtain eigenvalues and eigenvectors based on a Hamiltonian simulation. It just needs one ancilla qubit and one Hamiltonian evolution operator, which may be implemented on existing devices. In addition, it can obtain the energy spectrum of a Hamiltonian and each eigenstate directly rather than just the ground-state energy. The previous work in Refs. [20,21,22,23,24] determined the energy spectrum and ground state of a two-qubit low-energy effective Hamiltonian of the water molecule. We design new multi-round processing to improve the efficiency of the QRT algorithm that reduces the complexity from $O(\Delta E/\epsilon^{2})$ to O(R/ϵ), with an eigenvalues range ΔE, the number of eigenvalues *R* and the accuracy ϵ. We utilize a nuclear magnetic resonance (NMR) four-qubit quantum simulator to solve the eigenproblem of a three-qubit effective water molecule Hamiltonian in a given energy range. Non-Hermitian matrices are more common in a broader range of fields, such as data compression and principal component analysis. Therefore, we also show its ability to determine the singular vectors of a simple non-Hermitian matrix as an example.

In this paper, we introduce the QRT-based algorithm and our improvement in Section 2, show the details and results of the experiments in Section 3, and analyze some details of this work in Section 4.

## 2. Materials and Methods

QRT is a common quantum phenomenon in which energy will transit from one system to another. It often happens in two systems with close eigenenergies, such as photons and atoms. In order to make the QRT algorithm easier to understand, we first introduce the algorithm of determining eigenstates of $H_{s}$ with known eigenvalues and then explain how to obtain eigenvalues in a given range through QRT.

The Hamiltonian of the QRT algorithm is constructed as
(1)$$H=\frac{1}{2}\omega \sigma_{z}⊗I+H_{T}+c\sigma_{x}⊗A,$$
where *I* is the identity operator and $\sigma_{x,z}$ are the Pauli matrices. The first term in the above equation is the Hamiltonian of the probe qubit where ω is its frequency. The second term $H_{T}$ is the Hamiltonian of the work qubits in subspace |1〉 of probe qubit, and the third term describes the interaction between probe qubit and work qubits with coupling strength *c* and interaction matrix *A*. The Hamiltonian $H_{T}$ is given by
(2)$$H_{T}=\omega_{0}\left|0\rangle \langle 0\right|⊗\left|\Phi \rangle \langle \Phi \right|+\left|1\rangle \langle 1\right|⊗H_{s},$$
where $\omega_{0}$ is a parameter set as a reference point to the energy $E_{i}$ of $H_{s}$. In the algorithm, we firstly set the probe qubit in |0〉 and the work qubits in a reference state |Φ〉, which means the initial state of the circuit is an eigenstate of $H_{T}$ with eigenvalue $\omega_{0}$. Considering the example of solving the ground state of the $H_{s}$, we set $\omega +\omega_{0}\approx E_{0}$. Simulating the Hamiltonian in Equation (Equation 1) with evolution time τ, we apply the first-order perturbation theory with small $c\ll |E_{0}|$, and the state of all qubits can be formulated as
(3)$$\begin{array}{cc} |\Psi (\tau )\rangle & =e^{-iH\tau }|0\rangle ⊗|\Phi \rangle \\ \; & =e^{i\phi_{1}}\sin\left(\frac{\Omega_{0}\tau }{2}\right)|1\rangle ⊗|\Psi_{0}\rangle \\ \; & +e^{i\phi_{0}}\cos\left(\frac{\Omega_{0}\tau }{2}\right)|0\rangle ⊗|\Phi \rangle , \end{array}$$
where $|\Psi_{0}\rangle$ is the ground state of $H_{s}$ and $\Omega_{0}=2c\left|\langle \Psi_{0}\left|A\right|\Phi \rangle \right|$. Here, $\phi_{1}$ and $\phi_{0}$ are phases that are inessential in this algorithm. By measuring the probe qubit, we can easily affirm the state of work qubits. If the evolution time $\tau =\frac{\pi }{2c\left|\langle \Psi_{0}\left|A\right|\Phi \rangle \right|}$, the final state will be $|1\rangle ⊗|\Psi_{0}\rangle$ determinately. The procedure to prepare the eigenstates by QRT is shown as follows.

(i) *Initialize the state to $|0\rangle ⊗|\Phi \rangle$*. It should be noted that a better guess state $|\Phi \rangle$ obtained from preprocessing, such as the tensor-network method, will shorten the evolution time τ to improve the efficiency.

(ii) *Dynamical evolution*. The Hamiltonian is shown in Equations (Equation 1) and (Equation 2). Appropriate evolution time will increase the probability of success in the next step.

(iii) *Measure the probe qubit*. If $\omega +\omega_{0}\approx E_{i}$, the measurement result will be $|1\rangle$ with a certain probability, and the state of work qubits would be the corresponding eigenstate $|\Psi_{i}\rangle$. When the measuring result is $|0\rangle$, the state of all qubits is $|0\rangle ⊗|\Phi \rangle$, then we should return to step (ii) and run again until the measurement result is $|1\rangle$.

Compared with the original work, it saves one qubit to do the same things [20]. Usually, we just have an approximate eigenenergy but not an exact one, then the off-resonance will happen, and the amplitude of Rabi oscillation will be $\sqrt{\frac{\Omega_{0}^{2}}{\Omega_{0}^{2}+\Delta E^{2}}}$ where $\Delta E=E_{0}-\left(\omega +\omega_{0}\right)$. In other words, it is also accessible to $|\Psi_{0}\rangle$ with a high probability when $\Delta E\leq \Omega_{0}$.

In the above example, we have assumed that the energy spectrum of $H_{s}$ is almost known. In the following, we show how to obtain the energy spectrum in a given range of energy. In the previous algorithm, the probability of probe qubit in |1〉 will depend on evolution time τ if $\omega +\omega_{0}$ approximates any eigenvalue $E_{i}$. Assuming that the energy range is from $E_{min}$ to $E_{max}$, we pick *N* equally spaced points $\omega_{k}\left(k=1,2,\ldots N\right)$ in this range with step $\Delta \omega =\frac{E_{max}-E_{min}}{N}$ and set $\omega +\omega_{0}=\omega_{k}$, respectively, then run the previous algorithm. By measuring the probability $P_{k}$ of probe qubit in |1〉 for each $\omega_{k}$, it is obvious that QRT will happen if $\omega_{k}\approx E_{i}$, and the probability of |1〉 will increase in these points. As mentioned above, off-resonance will happen, and the amplitude of Rabi oscillation will be bigger when $\omega_{k}$ gets closer to $E_{i}$. We can find $\omega_{k}$ corresponding to the points of peaks (the approximate value $\tilde{E_{i}}$ of eigenvalue $E_{i}$) with a higher $P_{k}$, and reset $\omega +\omega_{0}$ near these points $\tilde{E_{i}}$ with smaller step Δω and *c* to reduce the error of off-resonance. The process that updates $\tilde{E_{i}}$ will repeat many times until the precision is sufficient, and the $\tilde{E_{i}}$ in the final process are the eigenvalues with limited error. It can be summarized by Algorithm 1. The original QRT method just set $\omega_{k}=E_{min}+k\epsilon$ where $k=0,1,2,\ldots \frac{E_{max}-E_{min}}{\epsilon }$. It will directly scan the interesting range of eigenvalues at regular intervals. By this loop program, the efficiency of the QRT-based eigenvalues search algorithm is improved due to ignoring non-resonance points.
**Algorithm 1:** Eigenvalues search by QRT**Inputs:**$H_{s}$, |Φ〉, $E_{min}$, $E_{max}$.*step 1:* Set $\omega_{k}=E_{min}+k\times \frac{E_{max}-E_{min}}{N}\;\left(k=0,1,\ldots N\right)$ and $c=c_{0}$.**Repeat the following steps:***step 2:* Set $\omega +\omega_{0}=\omega_{k}$.*step 3:* Run eigenstate search algorithm for each *k* with stepsize $\Delta \omega_{n}$.*step 4:* Obtain $P_{k}$ for each *k*.*step 5:* Update $\tilde{E_{i}}$.*step 6:* Set $\omega_{k}$ from $\tilde{E_{i}}-m\Delta \omega_{n+1}$ to $\tilde{E_{i}}+m\Delta \omega_{n+1}$, smaller $c_{n+1}$ and $\Delta \omega_{n+1}=c_{n+1}$.**Outputs:**$\tilde{E_{i}}$.

It is worth mentioning that a better initial state $|\Phi \rangle$ with an appropriate transition operator *A* will also shorten evolution time τ. For example, if we have an approximate guess state |Φ〉 about the target eigenstate (such as ground state), then we set A=I, and the evolution time would be greatly reduced to improve the efficiency.

## 3. Results

We demonstrate the algorithm in a four-qubit liquid nuclear magnetic resonance (NMR) system. In these experiments, the four-qubit sample used is ^13^C-labeled transcrotonic acid dissolved in d6-acetone with the ^1^H decoupled throughout the entire process. In this letter, the energies and time are recorded in units of Hartree and Hartree${}^{-1}$. The structure and parameters of this molecule are illustrated in Figure 1. Here, C1 is chosen as the probe qubit and C2-C4 are the work qubits. The internal Hamiltonian under a weak coupling approximation is
(4)$$H=-\Sigma_{i=1}^{4}\pi v_{i}\sigma_{z}^{i}+\Sigma_{i<j}^{4}\frac{\pi }{2}J_{ij}\sigma_{z}^{i}\sigma_{z}^{j},$$
where $v_{i}$ is the chemical shift and $J_{ij}$ is the *J*-coupling strength between the *i*th and *j*th nuclei. All experiments are carried out on the Bruker DRX 600-MHz spectrometer at room temperature (298 K).

### 3.1. Eigenvalues and Eigenstates of Water Molecule Hamiltonian

In this subsection, we demonstrate how to obtain the eigenvalues and determine the eigenstates by the QRT algorithm in a four-qubit NMR system. The schematic diagram and quantum circuit are shown in Figure 2. The first step of the NMR experiments is preparing the Pseudopure State (PPS). We use the spatial averaging technique method [25,26,27] to obtain it by several gradient fields and unitary operators. The density matrix of a four-qubit PPS is
(5)$$|\rho_{0000}\rangle =\frac{1-\epsilon }{16}I_{16}+\epsilon |0000\rangle \langle 0000|$$
where the polarization $\epsilon \approx 10^{-5}$. The identity matrix will not show any signal so the second term is the effective density matrix. In this algorithm, the initial state |0〉⊗|000〉 is the same with the effective density matrix of the PPS so we are actually done initializing the state after preparing the PPS. The density matrix of the PPS is constructed by using the quantum state tomography technology [28,29,30,31,32] and obtaining a fidelity of 99.51% between the experiment result and |0000〉〈0000|. The fidelity of the initial state is high enough that we can proceed to the next step.

The Hamiltonian simulation is the vital step in this algorithm. As mentioned above, the Hamiltonian we simulate is shown in Equations (Equation 1) and (Equation 2), where the quantum circuit to implement the evolution operator $U=e^{-iHt}$ using the Trotter formula is shown in Figure 2b. The state |Φ〉 is |000〉 here, and $H_{s}$ is the effective Hamiltonian of the water molecule in an eight-dimensional Hilbert space. The transition operator $A=H_{d}⊗H_{d}⊗H_{d}$, where $H_{d}$ is the Hadamard operator. Here, we set ω=1, and the details of the water molecule Hamiltonian $H_{s}$ are shown in Appendix A. Assuming that the interesting range of eigenvalues is from $-84.30\;(E_{min})$ to $-80.70\;(E_{max})$, we firstly set sixty points where $\omega +\omega_{0}$ is changed from $E_{min}$ to $E_{max}$ at regular intervals and c=0.05. To detect the energy spectrum of the Hamiltonian, we need to obtain the possibility of C1 (probe qubit) in |0〉, i.e., Tr(ρ|0〉〈0|III). By applying the readout pulse YIII, where *Y* denotes a rotation of π/2 along the *y* axis, $Tr(\rho |0\rangle \langle 0|III)=Tr(\rho \left(IIII+\sigma_{z}III\right)/2)$ can be extracted from the fitting of the experiment’s data.

The results of these experiments are shown in Figure 3a, which has some peaks in different coordinates. From the position of these peaks, we deduce eight peaks roughly corresponding to the eight eigenvalues of $H_{s}$: {−81.07 −81.98 −82.41 −82.65 −82.77 −83.02 −83.38 −83.93}. To improve the accuracy, we reset c=0.012 and rerun the algorithm with setting $\omega +\omega_{0}$ near the eight peaks. The results are shown in Figure 3b. They help us update the previous eight eigenvalues: {−81.04 −81.98 −82.44 −82.64 −82.75 −83.00 −83.38 −83.96}. Compared with the results of the numerical calculation, {−81.0447 −81.9802 −82.4325 −82.6418 −82.7594 −82.9918 −83.3756 −83.9558}, the difference values between the experimental and theoretical values are smaller than step length Δω.

When we obtain the energy spectrum of the Hamiltonian $H_{s}$ by our method or other methods, we can prepare the eigenstates of this Hamiltonian. Here, we assume that the accurate energy of the ground state is known and set $\omega +\omega_{0}=-83.9558$, which is the lowest energy level of $H_{s}$. After repeating the previous three steps, the final quantum state is $\left|1\rangle ⊗\right|\Psi_{0}\rangle$, with $|\Psi_{0}\rangle$ as the ground state of $H_{s}$. Usually, it is an entangled state with an arbitrary evolution time τ, and we need to measure the probe qubit until the measurement result is |1〉.

For the other eigenstates, if the measurement result of the probe qubit is |1〉, the state of the work qubits is the target state $|\Psi_{i}\rangle$ corresponding to the energy level $E_{i}\approx \omega +\omega_{0}$ similarly.

As an efficient demonstration of the algorithm, we reconstruct the density matrix of the ground state corresponding to the first peak $E_{a}$ in Figure 3b. We can obtain the ground state in the subspace of the probe qubit being in the state |1〉. To determine the eigenstates, we use quantum state tomography to rebuild the density matrix or quantum state with a series of readout pulses [28,29,30,31,32]. The ground state is represented in Figure 4a, and the fidelity is up to 98.66%.

### 3.2. Singular-Value Decompositions

The matrices are often non-Hermitian or non-square in many fields, where the singular-value decomposition is more important in a matrix analysis. For a matrix *M*, the SVD can be represented as this equation:(6)$$M=\sum_{i=1}^{N}s_{i}u_{i}{v_{i}}^{†},$$

*N* is the rank of matrix *M*, $u_{i}$ and $v_{i}$ are the *i*th left and right singular vectors corresponding to the singular value $s_{i}>0$.

*M* is a non-Hermitian matrix, and we can construct a new Hermitian matrix *B* [33]:(7)$$B=\left[\begin{array}{cc} 0 & M\\ M^{†} & 0 \end{array}\right],$$

0 is the zero matrix. *B* has the full information of matrix *M*, and we can obtain singular values and vectors by solving the eigenproblem of Hermitian matrix *B*. The *i*th eigenvalue and eigenvector of *B* are $s_{i}$ and ${\left[\begin{array}{cc} u_{i} & v_{i} \end{array}\right]}^{T}$. That is to say, if $H_{s}=\left[\begin{array}{cc} 0 & M\\ M^{†} & 0 \end{array}\right]$, by our algorithm, the final state in the work qubits is:(8)$$|\Psi_{i}\rangle =|0\rangle |u_{i}\rangle +|1\rangle |v_{i}\rangle .$$

It should be noted that the eigenvalues of *B* are $\left\{s_{0},s_{1}\cdots s_{N},-s_{N},\cdots ,-s_{1},-s_{0}\right\}$. For the eigenvalue $-s_{i}$, the eigenvector is ${\left[\begin{array}{cc} u_{i} & -v_{i} \end{array}\right]}^{T}$, which is the same except a minus. So, we do not need to set our parameter $\omega +\omega_{0}<0$ because of their same singular vectors.
(9)$$M=\left[\begin{array}{cccc} 3 & 0 & 0 & 0\\ 0 & 1 & 0 & 1\\ 0 & 1 & 1 & 0\\ 0 & 0 & 0 & 1 \end{array}\right]$$

Here, we use a non-Hermitian matrix *M* and construct a new Hermitian matrix *B* by Equation (Equation 7). We often pay more attention to the singular vectors corresponding to the larger singular values, such as in recommendation systems. The experiment details of solving singular-value problems are similar to Section 3.1, which are omitted here. Figure 5 shows the reconstructed density matrix of the left and right singular vectors $|\Psi_{0}\rangle =\frac{1}{\sqrt{2}}\left|0\rangle \right|u_{0}\rangle +\frac{1}{\sqrt{2}}\left|1\rangle \right|v_{0}\rangle$ of the maximum singular value, and the fidelity is 98.44%.

## 4. Discussion

The results of the original algorithm are similar to Figure 3a. It sets ω from $E_{min}-\omega_{0}$ to $E_{max}-\omega_{0}$ with the same stepsize ϵ; thus, the accuracy of the eigenvalues is ϵ obviously. The evolution time is t=1/c=O(1/ϵ) in each point, so the total complexity about the accuracy is $O(\Delta E/\epsilon^{2})$ where $\Delta E=E_{max}-E_{min}$. For our multi-round procedure, Equation (Equation 3) and the previous works [6,21] show that the widths of the resonant peaks are proportional to *c*. Therefore, the number of points near each eigenvalue is independent of $c_{l}$. It is the width/stepsize $\propto c_{l}/\Delta \omega_{l}=O\left(1\right)$. If we set $c_{l+1}=0.5\times c_{l}$, the total complexity with ϵ is O(1/ϵ) because $\sum_{l=1}^{l=L}1/c_{l}\leq 2/c_{L}=O\left(1/\epsilon \right)$. Here, we ignore the first round because $1/\epsilon_{0}\ll 1/\epsilon_{L}$. The complexity of this multi-round QRT algorithm is O(R/ϵ). If we focus only on a small number of eigenvalues with a high accuracy, such as the ground-state energy, the total time of the experiment will be greatly reduced. Compared with the other quantum algorithms, our improved QRT algorithm can obtain the arbitrary eigenstate of a Hamiltonian by a time evolution operator. Using the QSP, we can obtain the *j*-th eigenvalue and prepare the eigenstate with a query complexity $O(\epsilon^{-1})$. It just needs an evolution operator $e^{-iH/c}$ and measures one ancilla qubit, which may be friendly to the quantum device. More details can be found in Appendix B.

In this experimental work, here is a simple comparison of the resource required by the two methods. Assuming that we can implement the evolution operator $e^{-iH}$, we roughly calculate the number of times that different methods need to use this unitary operator. The repeating times of each point to obtain $P_{|1\rangle }$ is the same for the two methods in the experiments so we ignore this factor in this computing. For the original QRT, we set c=0.01 for each point. The number of points is ΔE/c=350. For each point, it needs the 1/c unitary operator $e^{-iH}$ to implement the Hamiltonian evolution $e^{-iHt}$ where t=1/c. So, the total number of the used operator $e^{-iH}$ is 35,000. For our multi-round method, c=0.05, so the total number of $e^{-iH}$ is 3.5/0.05×1/0.05=1400 where $c_{1}=0.05$. For the second round, it is $72\times 1/c_{2}=6000$. So, the total number of two rounds is 7400. Compared with the original QRT, our method just uses about 20% of the number of the operator $e^{-iH}$ to obtain the eigenvalues with the same accuracy. This method is not limited to two rounds, and the ratio can be further improved with more rounds.

The total running time of each experiment is about 20 ms, where T2 is about 1s for ^13^C so that we can ignore its influence. There are two factors resulting in the deviations of the final experimental states from the expected eigenstates: the fluctuations of the strength of the NMR signal and the error from the gradient ascent pulse engineering (GRAPE) pulses. The first part of the error is due to the difference between the ideal and experimental electromagnetic field strength, which includes the uncontrollable fluctuation of the experimental instruments and some systematic errors, such as the error generated by measuring the π pulse. For the second part, all the operators are optimized by GRAPE [28,34] in this four-qubit experiment, which is theoretically not perfect. The fidelities of the single-qubit gates and two-qubit gates that we set here are 99.99%, and the fidelities of the time evolution operators in the second step are 99.5%. Comparing the fidelity of the final states with PPS, our experiments confirm the validity of this quantum algorithm in the range of the errors permitted. In these experiments, two factors influence the scaling of the algorithm: the uncertainty ϵ of the eigenvalues and the evolution time of the Hamiltonian simulation. For the second part, some alternative methods such as the Trotter formula [35,36], sparse matrix [14,37] and quantum signal processing (QSP) [38] can be used to simulate a Hamiltonian efficiently. Although we cannot give specific scaling laws for simulating a realistic system, the computational resources required scale polynomially with the system size, giving a quantum speedup compared to classical computers [20,39].

This quantum algorithm needs to measure the probability of the state of one qubit. Recently, some quantum systems like the superconducting quantum circuits and nitrogen vacancy (NV) center have been promising to achieve practical computations beyond the classical computer. It is convenient and fast to achieve an ensemble measurement of one qubit in these quantum systems, making it more efficient to obtain the eigenvalues of a Hamiltonian by this algorithm.

## 5. Conclusions

To summarize, we demonstrated the optimized algorithm based on QRT to solve the eigenproblem of the three-qubit effective Hamiltonian of the water molecule without having a good initial state. This Hamiltonian’s energy and ground state are obtained in a four-qubit NMR quantum simulator with high fidelity. Using the multi-round method, we reduce the complexity from $O(\Delta E/\epsilon^{2})$ to O(R/ϵ). It is suitable for searching a small number of eigenvalues with high precision over a large range. Our algorithm can achieve a quadratic speedup over the phase estimation algorithm [22] and may perform better than adiabatic quantum computing in some problems [6]. Meanwhile, we also prepared singular vectors of a simple non-Hermitian matrix, showing the ability to solve singular-value decomposition problems. Our improved QRT algorithm reduces the quantum resource requirement for solving eigenproblems and provides an alternative for exploring the potential of quantum computers for matrix problems.

## Figures and Tables

**Figure 1 entropy-25-00061-f001:**
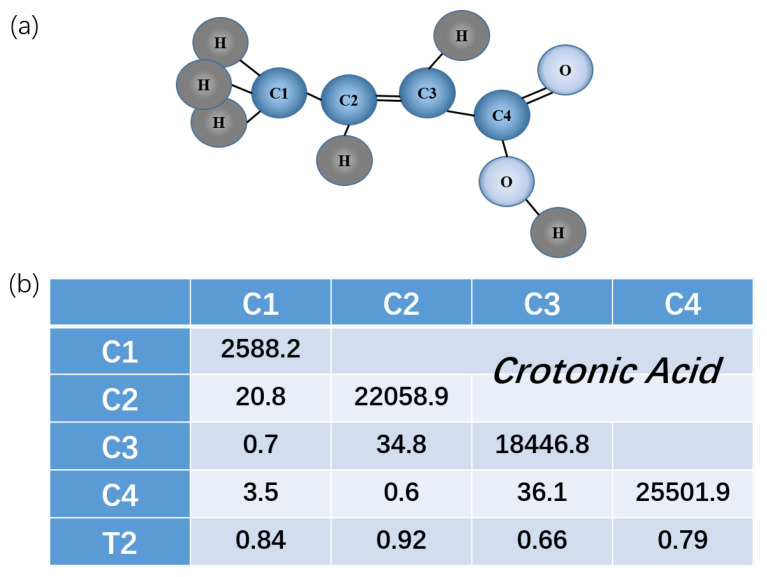
(**a**) Molecule structure of ^13^C-labeled crotonic acid. (**b**) Molecule parameters of ^13^C-labeled crotonic acid. C1 is probe qubit and C2-C4 are work qubits. ^1^H’s are decoupled throughout the experiment. Diagonal and non-diagonal elements are the chemical shifts and *J* couplings (in megahertz). Decoherence time T2s (in seconds) for each qubit are shown at the bottom.

**Figure 2 entropy-25-00061-f002:**
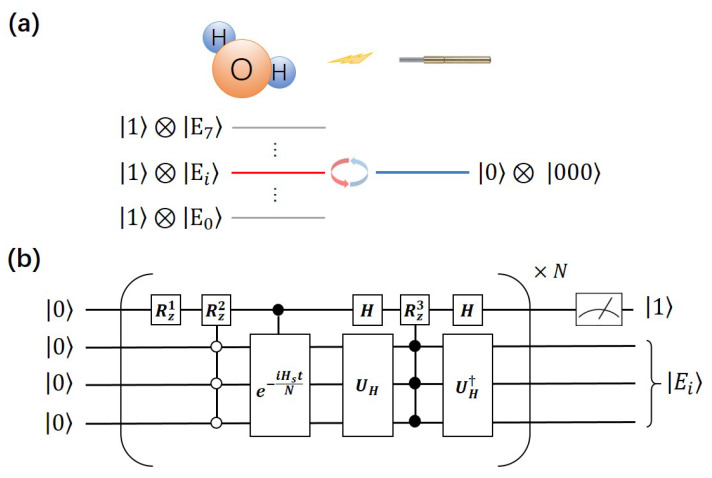
(**a**) The schematic diagram of this experiment. The left is water molecule and its eigenenergy where right is probe system. (**b**) The quantum circuit of the experiments using the Trotter formula. We set N=1/c here. The rotation angle of $R_{z}^{1},R_{z}^{2}$ and $R_{z}^{3}$ are $\frac{t}{N},\frac{\omega_{0}t}{N},\frac{2ct}{N}$. $U_{H3}$ is the Kronecker product of three single-bit operators $U_{H}=U_{H1}^{⊗3}$ that satisfies $U_{H1}\sigma_{z}U_{H1}^{†}=H_{d}$.

**Figure 3 entropy-25-00061-f003:**
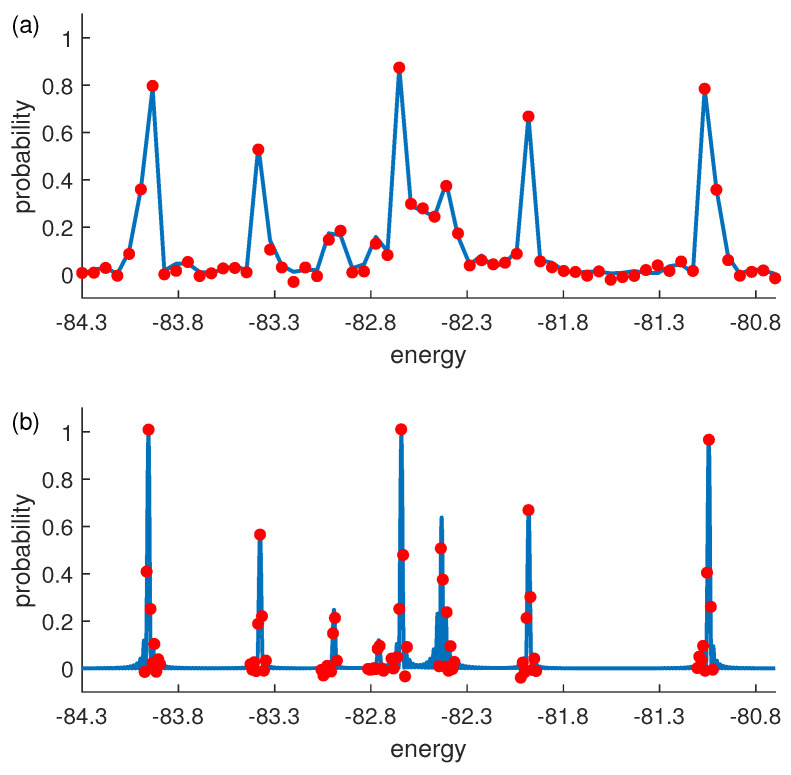
The probabilities of state |1〉 of probe qubit. Red points are experiment results and blue line is obtained by numerical simulation. (**a**) c=0.05. (**b**) c=0.012. Eight peaks represent eight eigenvalues of $H_{s}$. The probability $P_{k}$ of |1〉 is increasing just when $\omega +\omega_{0}\approx E_{i}$, and we are able to deduce the eigenvalues of $H_{s}$ by measuring probe qubit.

**Figure 4 entropy-25-00061-f004:**
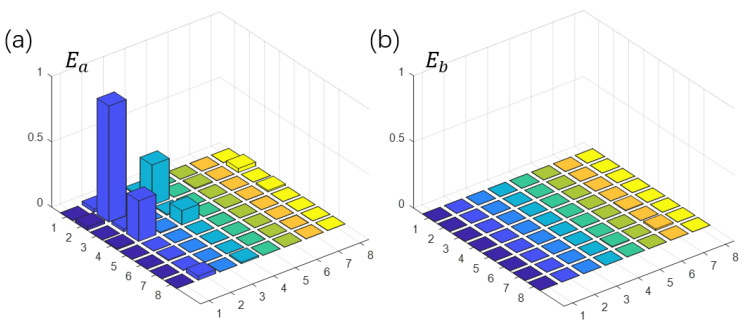
Experimental state tomography of the quantum simulator. (**a**,**b**) show the real parts of experimental reconstructed density matrices, and the rows and columns labeled 1–8 represent computational basis states from |1000〉 to |1111〉. (**a**) The ground state $|\Psi_{0}\rangle$ of $H_{s}$ corresponding to $E_{a}$ in Figure 3. (**b**) State of $E_{b}$ which is far from resonant point. We just show the elements of subspace where probe qubit is |1〉 because the other elements give no information about $H_{s}$.

**Figure 5 entropy-25-00061-f005:**
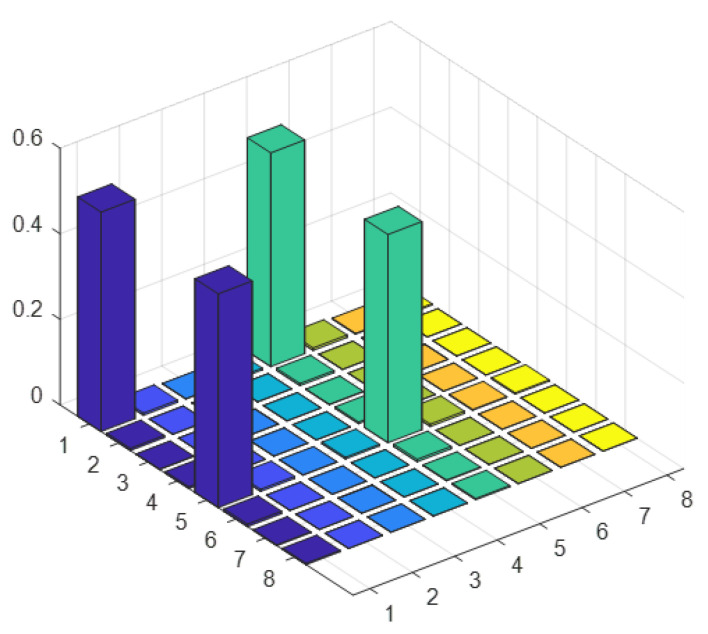
Experimental state tomography of the left and right singular vectors of *M* corresponding to maximum singular value. The rows and columns labeled 1–8 represent computational basis states from |1000〉 to |1111〉.

## Data Availability

The data presented in this study are available on request from the corresponding author.

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
