# Peer review of "Quantum Multi-Round Resonant Transition Algorithm"

_entropy, 2022, doi:10.3390/e25010061_

Round 1

Reviewer 1 Report

Summary:

This paper uses the concepts developed in the recent work by Jiangfeng Du and co-workers titled ‘Quantum Simulation of Resonant Transitions for Solving the Eigenproblem of an Effective Water Hamiltonian’ to calculate eigenvalues and eigenvectors of the effective molecular Hamiltinian  of water. The quantum resonant transition algorithm (QRT) developed in the earlier work is based on the simulation of the Hamiltonian where one starts from a simple reference state and carries out a quantum simulation of the resonance transitions from the initial state to the eigenstates of the target physical Hamiltonian. Jiangfeng Du and co-workers claimed that unlike the quantum phase estimation (QPE) algorithm, the QRT formalism does not require preparing an initial state with a good overlap with the final state. The present authors offer improvements over the earlier approach by a) reducing the number of qubits required by 1 and b) arguing that the complexity of the QRT algorithm comes out to be O(1/epsilon) where epsilon is the desired accuracy, in their multi-rounding scheme where the coupling strength between the probe qubit and the quantum state is changed in an adaptive fashion to refine the eigenvalues and consequently the eigenstates of the molecular Hamiltonian. It should be noted that the previous work mentioned that QRT scales as O(1/epsilon2). The authors make use of a NMR based quantum simulator in this work and illustarte the importance of the QRT algorithm for singular value decomposition applications as well.

Here are some of my observations:

1. "Quantum Phase Estimation Algorithm (PEA) [2,3] is one of the  most famous quantum algorithms, which can obtain the eigenvalues of a Hamiltonian with exponential speedup over classical algorithms" -> One should be careful about mentioning 'exponential speed up' —> kindly take a look at this recent paper "https://arxiv.org/abs/2208.02199" which mentions that exponential speed-up might not be guaranteed for generic chemical systems. One can maybe mention quantum advantage instead of exponential advantage.

2. Have the authors considered testing the QRT algorithm for strong correlation problems. This will test the efficiency of the algorithm when near -degeneracies are involved. For example, one can take the molecular Hamiltonian for a stretched geometry of the water molecule. Weak correlation problems like shown in this work, are handled quite well by the classical quantum chemistry algorithms as well.

3. The prefactor of the QRT algorithm might still be quite high even if scaling has been shown to be technically 1/epsilon. This could be specially true when chemical accuracy in higher energy excited states is desired. Can the authors comment on this aspect?

4. "Adiabatic Quantum Computing [4] is another quantum algorithm for preparing an eigenstate of a system, but scaling the algorithm’s run time remains an open question if the ground state is degenerate" —> kindly add a reference to support this claim.

5. "Almost all algorithms based on variation need many samples to obtain the estimated values of the different …'" —> maybe replace 'variation' by 'the variational principle'.

6. 'It can be summarized by Algorithm ????' -> typo!

7. "Diagnal and non-diagnal elements are the chemical shifts and J couplings (in megahertz). T2’s (in seconds) are shown at the bottom " -> define T2, typo in 'diagnal'.

8. UH3 not shown in the figure 2 but mentioned in the caption!

9. In figs 4 and 5 -> explain the axis, typo in ‘piont’

10. "Although we cannot give specific scaling laws for simulating a realistic system, the computational resources required scale polynomially with system size, giving exponential speedup than classical computers" —> refer again to point 1.

11. "The optimized algorithm uses fewer qubits than the original, which helps us solve a similar problem for a large Hamiltonian with the same quantum processor" —> only one qubit is reduced compared to the earlier work. So, this statement seems a little too much!

Reviewer 2 Report

This paper focuses on improving and testing a previously proposed algorithm, the quantum resonant transition (QRT) algorithm, for eigenvalue and eigenstate problem. Numerical tests verify the effectiveness of the improved QRT algorithm, and it is claimed that the improved algorithm can achieve a quadratic speedup compared to the original version in precision. 

The topic of eigenvalue problem is crucial in quantum information and is worth investigating. This paper is overall well organized, and the numerical results seem clear and reasonable to me. However, unfortunately, I am afraid that the current version of this paper has several critical issues. 

The motivation of this work is not clear to me. In particular, although solving eigenvalue problem is indeed important, it is not clear to me why studying QRT algorithm is needed. There are many other existing algorithms which can achieve 1/eps scalings (see e.g., algorithms in Table 1 of [arXiv:2204.05955]), and I do not see advantage of QRT algorithm compared to existing algorithms. I think it is necessary to provide a justification of the reason for studying QRT method and the aspect where QRT can be better than other algorithms. 

Related to my first concern, the literature review in the introduction might be inadequate. Many existing algorithms are not mentioned and compared. A thorough review and comparison with existing algorithms for eigenvalue problems is necessary but missing. 

The novelty of this paper is not clear to me. In particular, without reading the original QRT paper, it is not clear to me in which step the algorithm is different from the original version, and this makes it hard to assess the novelty of this paper. I think it might be helpful to add an overview of the original QRT algorithm, and explicitly highlight the difference (of the procedure of the algorithms, not only the complexity scalings) between the newly proposed algorithm in this paper and the original one. 

The main conclusion of this paper is not scientifically valid to me without further evidence. In particular, it is claimed that the improved algorithm can achieve 1/eps complexity scaling, but the proof in the paper (p.8, Section Discussion) is not rigorous. It seems that only the time in the Hamiltonian simulation subroutine is estimated, but this is not the complexity: there are many other subroutines which might introduce extra error but be ignored in the paper. For example, the usage of perturbation theory in Eq.(3) can introduce extra errors due to the ignorance of higher order terms. The step of estimating the probability P_k (I suppose this should be similar to the amplitude estimate and can have a worse scaling in eps) can have errors. Hamiltonian simulation step can also have errors, especially in this paper the Trotter method is used (mentioned in p.5), and the overall complexity of p-th order Trotter is T^{1+1/p}/eps^{1/p}, which is never linear in T and has extra factor in 1/eps (this flaw can be improved by using QSP for Hamiltonian simulation). Furthermore, there is no numerical results investigating the numerical complexity versus 1/eps. Therefore, before a rigorous and step-to-step error analysis and complexity estimate is provided, the claimed speedup is not scientifically reasonable. 

For these reasons, I cannot recommend acceptance of this paper in its current version. However, I think this paper has the potential to meet the journal’s criteria and the authors might be able to resolve my concerns. Therefore, I would like to recommend a major revision and reconsideration. The final outcome is unclear to me yet at this point. 

Here are some more comments that might be relevant: 

 - p.2, first paragraph, a formatting issue: [7–9,9–11]

 - p.2, it would be good to provide a derivation of Eq.(3). 

 - p.2, ``Considering the example of solving the ground state of the H_s , we set eps + eps_0 \approx E_0.’’ Please be specific and quantitative on this approximation. How close this approximation should be? How will this affect the overall error? 

 - p.3, ``By measuring the probability P_k of probe qubit in |1> for each eps_k’’. Please be specific here about the procedure of getting the probability and the number of the repeats to get the probability within tolerated error. 

 - p.3, a formatting issue: ``It can be summarized by Algorithm ??.’’

 - p.5,  ``using the Trotter formula’’. How large is the Trotter error? How large the N should be? 

 - p.8, ``In this experimental work,......the ratio can be further improved with more rounds.'' Please report and compare the actual numerical costs (e.g., the number of gates, the number of operators to achieve similar numerical errors) of different methods, not the estimated ones. 

Round 2

Reviewer 1 Report

The authors have incorporated the recommended changes in the manuscript.